# The Characteristics of Care Provided to Population(s) in Precarious Situations in 2015. A Preliminary Study on the Universal Health Cover in France

**DOI:** 10.3390/ijerph17093305

**Published:** 2020-05-09

**Authors:** François Birault, Stéphanie Mignot, Nicole Caunes, Philippe Boutin, Emilie Bouquet, Marie-Christine Pérault-Pochat, Bérangère Thirioux

**Affiliations:** 1Département de Médecine Générale, Faculté de Médecine et de Pharmacie, Université de Poitiers, 86000 Poitiers, France; stephanie.mignot@univ-poitiers.fr (S.M.); dr.caunes@wanadoo.fr (N.C.); ph.boutin@icloud.com (P.B.); 2Maison de Santé Pluriprofessionnelle des Couronneries, 115 r des Couronneries, 86000 Poitiers, France; 3Service de Gynécologie Obstétrique, Université, CHU de Poitiers, 86000 Poitiers, France; 4Service de Pharmacologie Clinique et Vigilances, Université, CHU de Poitiers, 86021 Poitiers, France; Emilie.BOUQUET@chu-poitiers.fr (E.B.); marie.christine.perault.pochat@univ-poitiers.fr (M.-C.P.-P.); 5Service de Pharmacologie Clinique et Vigilances, INSERM U1084-LNEC/INSERM CIC1402, Université, CHU de Poitiers, 86021 Poitiers, France; 6Unité de Recherche Clinique Intersectionnelle en Psychiatrie à Vocation Régionale Pierre Deniker, Centre Hospitalier Henri Laborit, 86021 Poitiers, France; berangere.thirioux@ch-poitiers.fr

**Keywords:** universal health cover, precarious populations, reimbursed drugs prescriptions, general practice

## Abstract

*Background*: The French Universal Health Cover (CMU) aims to compensate for inequalities between precarious and non-precarious populations, enabling the former to access to free healthcare. These measures rely on the principle that precarious populations’ health improves if healthcare is free. We designed a study to examine whether CMU fails to compensate for inequalities in reimbursed drugs prescriptions in precarious populations. *Material and method*: This retrospective pharmaco-epidemiological study compared the Defined Daily Dose relative to different reimbursed drugs prescribed by general practitioners (GPs) to precarious and non-precarious patients in France in 2015. Data were analysed using Mann–Whitney tests. *Findings*: 6 out of 20 molecules were significantly under-reimbursed in precarious populations. 2 were over-reimbursed. The 12 remaining molecules did not differ between groups. *Interpretation*: The under-reimbursement of atorvastatin, rosuvastatin, tamsulosine and timolol reflects well-documented epidemiological differences between these populations. In contrast, the equal reimbursement of amoxicillin, pyostacine, ivermectin, salbutamol and tiopropium is likely an effect of lack of compensation for inequalities. Precarious patients are more affected by diseases that these molecules target (e.g., chronic bronchitis, bacterial pneumonia, cutaneous infections). This could also be the case for the equal and under-reimbursement of insulin glargine and metformin (targeting diabetes), respectively, although this has to be considered with caution. In conclusion, the French free healthcare cover does not fail to compensate for all but only for some selective inequalities in access to reimbursed drugs prescriptions. These results are discussed with respect to the interaction of the doctor–patient relationship and the holistic nature of primary care, potentially triggering burnout and empathy decrease and negatively impacting the quality of care in precarious populations.

## 1. Introduction

In France, policies to control expenditure on medication are based on economic models which theoretically aim to improve efficacy, and to reduce inappropriate prescriptions, overconsumption and iatrogenesis [1,2]. However, these economic models-based policies practically limit patients’ access to necessary or essential drugs, and favour the use of less efficacious drugs, or drugs that are more likely to induce adverse reactions [3]. Whereas these policies surely create savings in the short-term, they have been posited to also trigger an increase in health costs in the long-term [2]. In addition, these measures (e.g., the use of less efficacious drugs and/or drugs inducing adverse reactions) have deleterious effects and harmful consequences, that may be significantly amplified in populations of precarious patients.

It is worth noting that the French Health System is complex (for more details, see Methods). Basically, people with a monthly income below EUR 720 for a single person, and EUR 1080 for a couple living in an urban area, are considered “precarious”. In contrast, when their monthly income is above this threshold (EUR 720/1080), people are considered “non-precarious”. Non-precarious people benefit from General Health Cover (in French: “Régime Général de l’Assurance Maladie” or RGAM). This concerns about 91% of the French population. These patients are 65% reimbursed by the RGAM for their medical expenses (consultations, pharmacological treatments, hospitalizations, exams, etc.). Thus, to be fully reimbursed, they have either to pay the 35% remaining costs (or co-pay), or to sign up to additional private insurance. In contrast, precarious people benefit from the so-called Universal Health Cover (in French: “Couverture Médicale Universelle” or CMU). In this case, precarious people are reimbursed up to 65% of their medical expenses by the RGAM, comparably to non-precarious people. However, their limited financial resources allow them to neither pay the 35% remaining costs, nor take out additional private insurance, in contrast to non-precarious patients. Consequently, the French Health Insurance System takes care of the 35% additional costs for CMU-beneficiaries, allowing them to be finally 100% reimbursed for their medical expenses (i.e., 65% + 35%).

CMU was introduced in 2000. However, important new rules were settled in 2015 to improve the way CMU functions. The year 2015 is, thus, a pivot date in the CMU history. By 31 December 2015, 8.21% of the population in France was concerned by this measure, amounting to 5.4 million people. Importantly, CMU-beneficiaries (precarious) in general are young, half are adults from 20 to 59 years of age, and 44% are under 20. These characteristics differ from the general French population, which is equally distributed across the four standard age brackets [4]. The representation of 60 years old and over is only very low. This is due to the fact that retirees benefit from another compensatory system, but also that their monthly incomes so far exceed the above-mentioned threshold [5]. There are also more female CMU-beneficiaries than RGAM-beneficiaries (non-precarious). In fact, in the 20–60-year bracket of the general population, percentages of men and women are equal. In contrast, women among CMU-beneficiaries account for 58% of the 20–40-year bracket, 54% of the 40–60-year bracket and 53% of the over 60-year-olds [6]. Moreover, for comparable age and gender, a strong prevalence of smokers is observed in CMU-beneficiaries in comparison to RGAM-beneficiaries (48.6% versus 29.4%) [6]. This also holds true for obesity (15.4% vs. 9.0%) [7]. Similarly, some medical conditions are more frequent in CMU-beneficiaries. For instance, there is a risk factor of 1.4 for terminal chronic kidney, cardiovascular or neurological diseases; 2 for epilepsy; 2.2 for diabetes; 2.4 for psychiatric diseases; and 4.7 for addictive disorders [6].

Furthermore, health expenditure is higher per CMU- than RGAM-beneficiary. Two main factors can explain this difference. Firstly, precarious populations tend to visit general practitioners [8,9] more frequently, thus generating greater expenses for medication [10]. Secondly, mortality [11] is higher, as is the incidence of chronic illnesses [6] in these populations.

These data, reporting significant differences relative to risk factors of diseases, incidence of chronic illnesses, mortality, but also life hygiene and high-risk behaviours between precarious and non-precarious populations, strengthen the need for adapted health insurance measures. According to the French government, precarious populations’ health improves if healthcare is free. This theoretical *parti pris*, which is not only interesting but fundamental to the French social principle of equality, motivated the development of the CMU-based measures. Accordingly, the French government assumed that CMU, allowing precarious populations to access free healthcare, compensates for inequalities between populations. However, to the best of our knowledge, no pharmaco-epidemiological study has so far examined this assumption. For that, we designed a new study based on the Defined Daily Dose (DDD), relative to different molecules that are prescribed by general practitioners (GPs). DDD, as a technical unit measurement, refers to the assumed average maintenance dose per day for a drug used for its main indications in adults (for more details, see Methods). Our working hypothesis was that Universal Health Cover (CMU) fails to compensate for inequalities in reimbursed drugs prescriptions in precarious populations. We considered DDD an interesting and promising way, although not exclusive, to evaluate this potential lack of compensation for inequalities. We further hypothesized that this lack of compensation is reflected in lower DDD for given molecules in precarious, compared to non-precarious, populations.

## 2. Materials and Methods

### 2.1. Study Period

This study focused on the year 2015 (see below “Data collection and analysis”).

### 2.2. Data Collection and Analysis

Data were obtained and collected from the French Statistic Institute for Private Practitioners (in French “Institut Statistique des Professionnels Libéraux” or ISPL). ISPL notably enables the aggregation of data, regarding drugs reimbursement, from the French Health Insurance System and insurance providers. We here note that permission to contact ISPL and to collect data from its database is only rarely obtained. Accordingly, we were not allowed to collect data relative to the patients’ age, gender, underlying medical conditions, etc. Moreover, we were allowed only to collect data from the year 2015. Thus, only data relative to yearly drugs reimbursement (in 2015) were extracted from the ISPL database.

#### 2.2.1. Data Selection Relative to Prescribers’ and Patients’ Populations

We focused our analyses on the yearly reimbursement rates of drugs that were prescribed in 2015 by GPs. This was done for two main reasons. Firstly, GPs prescribe to all the French population, regardless of gender. As an example, this is not the case for urologists, gynaecologists or midwives [12]. In addition, the latter have law-limited prescriptions. Secondly, we focused on primary care and not on medical care specialties, as French patients have to consult their GPs in order to obtain permission to consult a specialist practitioner and to be reimbursed for this consultation. Of course, French patients can consult directly a specialist practitioner without obtaining permission from their GPs, but in this case they are not reimbursed for their medical expenses by the French Health Insurance System. Our analyses focused on the RGAM-beneficiaries and CMU-beneficiaries. The French Health Insurance System entails four different insurance covers: (1) The General Health Cover (or RGAM; see Introduction); (2) the Alsace–Moselle Health Cover (in French: “Régime Alsace–Moselle” or RAM); (3) the Universal Health Cover (or CMU; see Introduction); and (4) the State Medical Aid (in French: “Aide Médicale d’Etat” or AME). Based on resources criteria (see Introduction), patients benefiting from the RGAM or RAM are considered non-precarious populations, whereas patients benefiting from the CMU or AME are considered precarious populations. RAM was settled in the immediate post-World War II period. It concerns a very low part of the French population and does not correspond to the standards laid down in the General Health Cover. AME only concerns irregular migrants. As such, it does not target precarious populations per se, even if most foreigners in irregular situations experience precarious life conditions. Accordingly, data from both RAM- and AME-beneficiaries were not included in the present study. We will now use “non-precarious group” (NP) and “precarious group” (P) to respectively refer to the RGAM- and CMU-beneficiaries.

#### 2.2.2. Data Selection Relative to Medications and Criterion of Principal Evaluation

For medications, we used the Anatomical Therapeutic Chemical (ATC) classification. The World Health Organization (WHO) Collaborating Centre for Drug Statistics Methodology is in charge of this classification. Medications are divided into different groups according to the organ or system on which they have an action, and according to their therapeutic and chemical features. This classification is based on five levels. Respectively, the 1st level (first letter) indicates the anatomical group (among 14 different ones); the 2nd level (first two numbers) the main pharmacological or therapeutic subgroups; the 3rd and 4th levels (second and third letters) the therapeutic, pharmacological or chemical subgroups; and the 5th level (last two numbers) the chemical substance. This classification allows a therapy to be apprehended via analysis of the pharmaceutical used. As an example, metformin is a chemical substance corresponding to the 5th level (defined by letters and numbers as follows: A10BA02), which belongs to the chemical group of biguanides (4th level, A10BA) representing the pharmacological family of hypoglycemic agents (3rd level, A10B), a component of the therapeutic family of drugs indicated in diabetes (2nd level, A10), belonging to the anatomical group of the digestive system (1st level, A) (Table 1).

Table 1 indicates the total amount of drug packs reimbursed and corresponding percentage rate, as well as the total reimbursement cost (in EUR) and corresponding percentage rate for each ATC class (1st level) in France in 2015. Totals for all classes considered collectively are also reported.

We used the Defined Daily Dose (DDD) as a criterion of principal evaluation. That is, we hypothesized that differences in reimbursed drugs prescriptions are reflected in DDD modulations for given molecules between populations.

Drugs consumption can be expressed in cost, number of units, number of prescriptions, or by the physical quantity of drugs. DDD is a technical unit of measurement and is determined by WHO. It refers to the assumed average maintenance dose per day for a drug used for its main indications in adults. DDDs are only assigned for medicines given with an ATC code. DDD is notified on each pack by the producer and corresponds, thus, to a daily cost. Applying DDD enables the testing of changes in drugs utilization over time, to compare and evaluate the effect of an intervention on drug use, to document the relative therapy intensity with various groups of drugs, to track changes in the use of a class of drugs, and to evaluate regulatory effects and effects of interventions on prescribing patterns [13].

Here, the expenditure rate was collected based on the DDD per patient over 20 years of age for each pharmaceutical form and for each selected molecule in 2015. For each ATC class, we selected two molecules, i.e., depending on two criteria. That is, the molecule with the highest total cost of reimbursement and the molecule with the highest amount of packs reimbursed. We used this methodology as some molecules are sold at a high price, i.e., in packs containing only low daily doses (i.e., for short-lasting treatments) (e.g., amoxicillin). In contrast, some molecules are sold at lower prices, i.e., in packs containing more daily doses (i.e., for long-lasting treatments) (e.g., atorvastatin). When a given molecule for a given ATC class responded equally to these two criteria (total cost of reimbursement and total amount of packs reimbursed), this was uniquely retained. For given ATC classes, there was, thus, only one molecule selected. We here note that the antineoplastic class L was excluded, as GPs are not the prime prescribers for this class.

### 2.3. Statistical Analyses

Statistical analyses were computed using Jamovi Software© (open source software, Jonathon Love, University of Newcastle, University Dr, Callaghan NSW 2308, Australia).

We firstly calculated the amount of reimbursed DDD for a pack per precarious patient (P) and non-precarious patient (NP). Secondly, we computed the mean DDD for each molecule and for all packs.

Thirdly, we tested for normality using the Shapiro–Wilk tests. For each molecule, the *p*-value was < 0.05, indicating a violation of the assumption of normality. Accordingly, we used Mann–Whitney tests to compare the two groups (precarious (P) vs. non-precarious (NP)).

## 3. Results

For four out of all 1st level ATC classes (except for the antineoplastic class L that was not included in our data), only one molecule equally responded to the two criteria of total cost of reimbursement and total amount of packs reimbursed. This concerned the H, M, N and P classes (Table 1 and Table 2).

In total, 20 molecules were identified. In average, the reimbursement rate for all these molecules when considered globally was lower in P (EUR 1.28) than in NP (EUR 7.32) (Table 2).

Table 2 indicates for each molecule in each ATC class (First Level) the amount of different packs reimbursed, the total reimbursement cost and its corresponding percentage rate, DDD (mean ± SD) for the non-precarious group (NP), DDD (mean ± SD) for the precarious group (P) and *p*-Values (Mann–Whitney test comparing NP vs. P). Significant differences are indicated in bold (last column). Non-precarious people are significantly more reimbursed than the precarious people for metformin (A Class), atorvastatin and rosuvastatin (C Class), tamsulosine (G Class), paracetamol (N Class) and tiomolol (S Class). In contrast, precarious people are more reimbursed than non-precarious people for econazole and ciclopirox (D Class) (see the data in red in the column DDD-P). Non-precarious tend to be more reimbursed for rivaroxaban (B Class) and salbutamol (R Class).

For 10 out of the 20 molecules, there was no significant difference between groups (all *p* > 0.05; Table 2). This was the case for insulin glargine (A class), acetylsalicylic acid (B class), *Serenoa repens* (G class), prednisolone (H class), amoxicillin and pyostacine (J class), ibuprofen (M class), ivermectin (P class), tiopropium (R class), and cromolyn sodium (S class). For two molecules, there was a trend to significance, i.e., rivaroxaban (B class) and salbutamol (R class) (Table 2). These two molecules tended to be less reimbursed in P than NP. The remaining eight molecules significantly differed between P and NP, i.e., metformin (A class), atorvastatin and rosuvastatin (C class), econazole and ciclopirox (D class), tamsulosine (G class), paracetamol (N class) and timolol (S class) (Table 2). Except for econazole and ciclopirox, which were more reimbursed in P than NP, all the remaining six molecules showed the inverse pattern (i.e., NP > P) (Table 2).

To sum up, our data suggest that precarious populations were significantly more reimbursed than non-precarious populations for molecules belonging to D class, targeting dermatological diseases (e.g., mycoses). In contrast, they were less reimbursed for molecules belonging to A, C, G, N and S classes, respectively targeting diseases that affect the metabolic, cardiovascular, genito-urinary, nervous and sensory organs systems (e.g., diabetes, dyslipidemia, pain, prostate adenoma, glaucoma, etc.).

## 4. Discussion

The efficacy of free healthcare cover in precarious populations needs to be addressed to evaluate the usefulness of public expenditure. For that, the present pharmaco-epidemiological study sought to test whether the French free healthcare cover, i.e., CMU, enables compensation for inequalities in reimbursed drug prescriptions in precarious populations. To the best of our knowledge, this is the first study that used a pharmacological criterion, i.e., DDD, to address this epidemiological question. Our working-hypothesis was that CMU fails to compensate for inequalities in reimbursed drugs prescriptions in precarious (CMU-beneficiaries (or P)) compared to non-precarious (RGAM-beneficiaries (or NP)) populations. More precisely, we assumed that this lack of compensation is reflected in lower DDD for given molecules in precarious- than non-precarious populations. With this aim, we compared the yearly reimbursement rates of drugs that were prescribed in 2015 by GPs to precarious and non-precarious patients. Our results partially confirmed our hypothesis. That is, 12 out of all 20 tested molecules were comparably reimbursed in both patients’ groups. Of these 12 molecules, 2 tended to be less reimbursed in precarious patients. The eight remaining molecules significantly differed between groups: six were significantly less reimbursed in precarious patients and two showed the inverse pattern. Consequently, our study suggests that the French free healthcare cover does not fail to compensate, for all but only for some, selective inequalities in access to reimbursed drugs prescriptions. More importantly, this lack of compensation was not only evidenced by under-reimbursed drugs in precarious compared to non-precarious populations (e.g., metformin and paracetamol); it was also evidenced by the absence of significant differences between groups regarding drugs that should have been over-reimbursed in precarious patients. These points are discussed below.

Although these results raise important questions regarding the French Health Insurance System and notably the free healthcare cover, there are nevertheless several limitations to the present study that need to be highlighted. Firstly, concerning our database, access to the ISPL database (see Methods) only provided aggregated data on amounts reimbursed per pack, regardless of the number of pills these contained. The French National Institute of Statistical Economic Studies (INSEE) collects, among others, data relative to the number of pills per pack. Thus, access to the INSEE database for the year 2015 would have certainly enabled computation of more refined data analyses. Secondly, we were only allowed to extract data relative to yearly drugs reimbursement (see Methods). That is, we had no access to data relative to the patients’ age, gender, health status and underlying medical conditions. Accordingly, we were not able to adjust our statistical analyses to these variables. Thus, no ponderation or stratification test has been performed. These are major confusion biases. In fact, it is well-documented (see Introduction) that there are significantly more women and younger people in precarious than non-precarious populations. Precarious populations also have higher risk factors to develop certain diseases (e.g., diabetes, renal, cardiovascular, neurological (e.g., epilepsy) and psychiatric disorders) [7]. These differences between populations probably impact, thus, the prescriptions of specific drugs. Thirdly, our study also entails an important selection bias. As we focused our analyses on the amounts of reimbursed drugs prescriptions, we were also not able to estimate the proportion of so-called irreducible inequalities, that is, the proportion of precarious patients who cannot access care because some practitioners refuse to provide consultation to precarious patients.

In general, our data show that differences in reimbursed molecules between groups differed according to ATC classes. We here start discussing the significant differences observed between precarious and non-precarious populations.

Regarding metmorfin, two possible explanations may be advanced, although with caution. Firstly, the prevalence of diabetes in the general population could be an epidemiological explanation for its under-reimbursement in precarious populations [14]. In fact, the average age of patients suffering from diabetes is ~63.1 years, with a gender ratio female/male of 1.04. Therefore, diabetes more frequently affects older populations. As already mentioned in the introduction, CMU is no longer allocated to over 65-year-olds. This could explain the lower expenditure observed in precarious patients. Thus, this effect would be due to epidemiological features of the non-precarious populations, but not to the CMU’s failure to compensate for inequalities. Secondly, the prevalence of diabetes, however, is well-documented to be higher in precarious populations [7]. If correct, we should have observed an over-reimbursement of metformin in our data, meaning, in contrast, that CMU does not compensate for inequalities. Therefore, because no stratification test was performed, it is difficult to explain this effect, which needs to be further investigated.

A comparable age-based reasoning applies to statins. Atorvastatin and rosuvastatin indications target hypercholesterolemia, which is more frequently observed in older people [15,16]. Similarly, the fact that there are significantly more young people as well as women in precarious populations could account for the over-reimbursement of tamsulosine in non-precarious patients in our data. Tamsulosine is used as an alpha-blocker, especially to treat the symptoms associated with benign prostates hypertrophy. As such, these are usually more prescribed to older men in the general population [17]. This holds also true for timolol. Timolol is indicated for cardiovascular diseases and open-angle glaucoma, which are known to more frequently affect older men and older people, respectively. Consequently, our data relative to statins, tamsulosine and timolol seem to be concordant with the literature [7].

In contrast, the prevalence and occurrence of pain and fever are not explained by age and gender in the general population. Thus, the different distribution in age and gender between precarious and non-precarious populations is unlikely to explain the over-reimbursement of paracetamol in non-precarious patients in our data, in accordance with previous studies [18].

Considered collectively, our results indicate that precarious patients were less reimbursed for molecules belonging to A, C, G, N and S classes, respectively targeting diseases that affect the metabolic, cardiovascular, genito-urinary, nervous and sensory organs systems (e.g., molecules indicated for diabetes, dyslipidemia, pain, prostate adenoma, glaucoma, etc.). However, these data reflect socio-demographical differences between groups, rather than a lack of compensation for inequalities. This is probably the case for all the above-quoted molecules, except for metformin and paracetamol. Future studies using stratification tests are needed to verify the impact of age and gender variables on this difference in reimbursed drugs prescriptions.

An over-reimbursement of econazole and ciclopirox (D Class), targeting dermatological diseases such as mycoses, was found in precarious patients. This over-prescription of antifungals has been well documented in the literature, as fungal infections are predominant among poor populations because of inadequate hygiene and unfavourable working conditions generated by poverty [19]. Moreover, we here posit that this effect could be also triggered by social desirability factors [20]. That is, non-precarious patients would feel more comfortable directly asking their chemist to provide them with antifungal drugs rather than consulting their GPs. In other words, non-precarious patients would prefer to pay for econazole and ciclopirox-based drugs without being reimbursed for their expenses, rather than show their GPs their body parts affected by mycoses. Because of their low financial resources, precarious people cannot allow themselves to act this way.

Of all the 20 tested molecules, 12 did not significantly differ between groups. The absence of difference between groups regarding amoxicillin, pyostacine, ivermectin, salbutamol and tiopropium is unexpected. We should have normally observed an over-reimbursement of these molecules in precarious populations based on epidemiological features. The data relative to insulin glargine, targeting diabetes, need to be considered with caution for the same reasons as the data relative to metformin, which was found to be over-reimbursed in non-precarious patients (see above). In contrast, the non-significant results concerning the remaining molecules were expected. In fact, indications of acetylsalicylic acid, rivaroxaban, *Serenoa repens*, prednisolone, ibuprofen and cromolyn sodium normally lead to homogenous prescriptions regardless of population (i.e., precarious or non-precarious). This is in line with our data.

Regarding amoxicillin, prescription has been shown to be independent from age [21]. This is concordant with our results showing that amoxicillin is comparably reimbursed in both precarious and non-precarious groups. However, WHO has stated that poverty is one of the main factors driving resistance to antimicrobials. In the United States, poverty-driven practices, such as sharing drugs, the use of “leftover” antibiotics, and the purchase and use of drugs of uncertain quality manufactured abroad, may be contributing to resistance to antimicrobials. Unfortunately, there are currently very few studies on the socio-economic and behavioral factors underpinning resistance to antimicrobials [22]. Nevertheless, the fight against the social determinants of poverty in the world remains a crucial yet neglected facet in the prevention of antimicrobial resistance [23]. Misuse could be one of the causes [24], such as self-medication—using leftovers in the family medicine cabinet. The fact that prescribers are not necessarily aware of the low levels of literacy in this population [25], leading to errors in observance, could be an explanation for these different consumptions. However, it seems that the resistance to antimicrobials precisely relates to the under-prescription of these same antimicrobials to poor populations. Consequently, this equal reimbursement of amoxicillin-based drugs between precarious and non-precarious patients may result from an under-prescription of antimicrobials to precarious populations, compared to the higher prevalence of infectious diseases that is encountered in these populations. This hypothesis is further reinforced by our data relative to pyostacine.

In fact, regarding pyostacine, our results were unexpected. In fact, its epidemiology tends to point towards over-prescription, and thus over-reimbursement, in precarious populations. The four therapeutic indications are sinusitis, obstructive chronic bronchitis, community-acquired respiratory diseases and cutaneous infections. Living in low-income geographical zones is associated with more serious chronic rhinosinusitis [26]. The prevalence of chronic bronchitis is linked to poverty, altitude, air pollution and policies on tobacco use [27]. Adults from underprivileged backgrounds run a greater risk of bacterial pneumonia [28] and cutaneous infections [29]. Comparably to amoxicillin, pyostacine-based drugs should have been more reimbursed in precarious compared to non-precarious populations.

The reimbursement of salbutamol and tiopropium was not found to significantly differ between groups. Similarly, this comparable reimbursement rate contrasts with the prevalence of obstructive respiratory tract disorders among poor populations, both at individual level and community level, in many countries [30]. Our results therefore concur with the literature for these molecules.

Considered collectively, these results indicate that the lack of compensation for inequalities in precarious populations is not only visible through under-reimbursement (metformin, paracetamol). It is also—if not more—visible through the absence of significant difference between groups regarding drugs that should have been over-reimbursed in precarious patients (i.e., amoxicillin, pyostacine, ivermectin, salbutamol and tiopropium). This further suggests that not only under-reimbursement, but also equality of reimbursement of given molecules, could be used as a suitable criterion to track for inequalities between precarious and non-precarious populations.

To sum up, the under-reimbursement of specific drugs (atorvastatin, rosuvastatin, tamsulosine and timolol) in precarious populations suggests a potential effect of the age and gender of non-precarious populations on the prevalence of specific diseases (dyslipidemia, glaucoma, prostates hypertrophy, cardiovascular diseases, etc.). It means that this under-reimbursement may be explained by epidemiology: certain pathologies are more frequent in men and older people, the percentage rates of which are higher in non-precarious than precarious populations. This hypothesis needs to be verified in future studies, (1) based on socio-demographical data, and (2) using stratification tests. This is necessary to confirm the impact of the age and gender variables on reimbursed drugs prescriptions. With the same aim, further studies are also needed to test for the impact of age on the prescription of metformin and insulin glargine in both precarious and non-precarious populations. In fact, the prevalence of diabetes significantly increases with age in the general population, but is also higher in precarious people. Thus, at this stage, it is not answered whether under-reimbursement of metformin in precarious populations, and equal reimbursement of insulin glargine, reflect epidemiological differences or the CMU failure to compensate for inequalities. In contrast, we here hypothesize that the equal reimbursement of amoxicillin, pyostacine, ivermectin, salbutamol and tiopropium could be a deleterious effect of lack of compensation for inequalities.

What explanatory model could best explain these results? One plausible hypothesis is that precarious patients do not seek care or therapeutic treatment because they are unaware of their disease due to socio-educational difficulties, triggering an incapability to recognize their symptoms and health status. For instance, lack of insight, i.e., unawareness of one’s mental illness, is frequently encountered in psychiatric conditions and has deleterious effects on prognosis, adherence to treatment, acceptance of hospitalisation and risk of relapse [31]. A comparable lack of awareness relative to somatic diseases (different from pure anosognosia, which is a core etiological element of given neurological pathologies) could also explain why precarious patients do not seek care. A second possible hypothesis, not very well documented in the literature, is the interaction of the doctor–patient relationship and the holistic nature of primary care [32]. First of all, the low rates of literacy and language difficulties are already an extra workload that could potentially exhaust practitioners in their doctor–patient relationship. It could be that practitioners favour long-established molecules with, a low level of interaction or few adverse reactions, so that they do not have to give any lengthy or complex explanations to their patients. Furthermore, the complexity of holistic care could also be an aggravating factor. Practitioners could decrease the prescription time to reduce the consultation time. Therefore, molecules with the simplest administration modes (i.e., daily, with no interaction with the bolus) could be preferred. Finally, when practitioners are exposed to a mainly precarious population, their exhaustion could reinforce the above-mentioned mechanisms.

Non-clinical factors come into play in the motivations for prescribing drugs [33]. A. Vega confirms the existence of several cultural tendencies, which are conducive to an accumulation of medications in prescriptions, and of which medical training courses and the organisation of treatment in France are the reflection. Regarding the different levels of prescription, they could depend on the level of professional fatigue (or burnout [32]), and on the initial motivations of each prescriber to become a doctor and practice in the salaried sector.

## 5. Conclusions

Precarious populations have limited access to care. France has chosen to try and reduce one of the factors limiting access to care by funding, in its entirety, through national solidarity, the cost of drugs. Our data, although still preliminary, suggest that this measure does not cancel out differences in reimbursed drug prescription for this population compared to the general population. Some drugs were indeed found to be less-reimbursed, and thus probably under-prescribed. Other drugs were equally reimbursed, and thus probably equally prescribed, in precarious and non-precarious populations, although they should have been over-reimbursed in the former. There is no justification in the literature explaining these observations. This could be due in part to the potential inefficacy of the financial assistance for accessing treatment in a fair, income-based system. A more specific approach (age, gender ratio, etc.) to these populations seems therefore necessary concerning prescriptions. The primary care practitioners’ holistic approach should be integrated, as well as the exhaustion they feel in their relationship with these populations, which can significantly decrease their empathy [34]. A study investigating the relationship between burnout in primary care practitioners, empathy decrease towards their precarious patients, and modulations of reimbursed drugs prescriptions, is already in progress in our laboratory.

## Figures and Tables

**Table 1 ijerph-17-03305-t001:** Anatomical Therapeutic Chemical (ATC) classification (1st Level) and total reimbursement cost in France in 2015.

ATC Classification—First Level	Amount of Drug Packs	%	Total Reimbursement Cost (€)	%
**A:** Alimentary tract and metabolism	286,733,847	16.5	1,828,493,537	19
**B:** Blood and blood forming organs	80,392,881	4.6	741,836,314	7.7
**C:** Cardiovascular system	229,121,391	13.2	2,398,652,845	24.9
**D:** Dermatologicals	56,592,966	3.3	159,282,939	1.7
**E:** Genito-urinary system and sex hormones	37,823,139	2.2	204,403,139	2.1
**H:** Systemic hormonal preparations, excluding sex hormones and insulins	53,694,282	3.1	164,408,842	1.7
**J:** Antiinfective for systemic use	105,779,412	6.1	843,586,059	8.8
**L:** Antineoplastic and immunomodulating agents	3,250,166	0.2	285,369,090	3
**M:** Musculo-skeletal system	99,171,188	5.7	301,048,270	3.1
**N:** Nervous system	627,153,800	36.1	1,649,956,726	17.2
**P:** Antiparasitic products, insecticides and repellents	4,175,031	0.2	27,614,468	0.3
**R:** Respiratory system	126,271,089	7.3	881,332,845	9.2
**S:** Sensory organs	24,540,639	1.4	79,069,409	0.8
**V:** Various	1,484,402	0.1	50,060,496	0.5
**Total**	**1,736,184,233**	**100**	**9,615,114,978**	**100**

**Table 2 ijerph-17-03305-t002:** Defined Daily Dose (DDD) for each molecule in each Anatomical Therapeutic Chemical (ATC) class in precarious and non-precarious groups.

Class	Molecule	Amount of Different Drug Packs Reimbursed	Total Reimbursement Cost		DDD-NP		DDD-P		*p*-Values
			(€)	%	Mean (€)	SD	Mean (€)	SD	
**A**	metformin	365	121,371,689	6.8	0.0993	0.324	0.0068	0.020	**<0.001**
	insulin glargine	10	250,044,490	14.1	0.5290	0.125	0.0028	0.006	0.140
**B**	acetylsalicylic acid	23	70,177,474	4.0	0.0559	0.098	0.0174	0.024	0.235
	rivaroxaban	18	158,306,679	8.9	0.0843	0.206	0.0048	0.022	0.056
**C**	atorvastatin	412	177,784,547	10	0.0553	0.150	0.0030	0.009	**<0.001**
	rosuvastatin	18	272,713,184	15.4	0.0989	1.440	0.0611	0.099	**0.015**
**D**	econazole	82	3,796,504	0.2	0.0003	0.000068	0.000094	0.002	**0.024**
	ciclopirox	30	9,459,732	0.5	0.0009	0.000039	0.0032	0.001	**0.005**
**G**	*Serenoa repens*	52	26,462,165	1.5	0.0185	0.060	0.0086	0.029	0.215
	tamsulosine	65	11,810,032	0.7	0.1720	0.027	0.0053	0.008	**0.004**
**H**	prednisolone	66	40,510,030	2.3	0.0326	0.067	0.0175	0.357	0.266
**J**	amoxicillin	280	61,215,883	3.4	0.0055	0.015	0.0045	0.010	0.681
	pyostacine	4	97,990,458	5.5	0.0856	0.150	0.0378	0.066	0.686
**M**	ibuprofen	117	32,875,248	1.9	0.0185	0.052	0.0153	0.044	0.420
**N**	paracetamol	257	274,135,679	15.4	0.0557	0.233	0.0288	0.140	**0.047**
**P**	ivermectin	4	14,682,048	0.8	0.0055	0.008	0.0095	0.017	0.486
**R**	salbutamol	39	46,967,969	2.6	0.3010	1.070	0.2410	0.883	0.053
	tiopropium	6	87,766,817	4.9	0.3000	0.479	0.0503	0.008	0.310
**S**	cromolyn sodium	25	5,440,587	0.3	0.0020	0.003	0.0063	0.103	0.130
	timolol	71	11,353,590	0.6	0.0253	0.066	0.0050	0.012	**<0.001**
**Total**			**1,774,864,805**	**100**					

Bold: Significant differences.

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
