# Peer review of "The Characteristics of Care Provided to Population(s) in Precarious Situations in 2015. A Preliminary Study on the Universal Health Cover in France"

_ijerph, 2020, doi:10.3390/ijerph17093305_

Round 1

Reviewer 1 Report

The quality of the manuscript entitled “The Characteristics of Care Provided to Population(s) in Precarious Situations in 2015. A Preliminary Study on the Universal Health Cover System in France” has significantly been improved in since the first version. However, some issues remain:

In the Abstract :

In the findings the authors state “6 out of 20 molecules were significantly under-reimbursed in precarious populations. 2 were over reimbursed.” This add up to 8 so how come the authors talk about the remaining 10 and not 12? “The 10 remaining molecules did not differ between groups.”

In the Introduction :

The 2 first sentences are inconsistent. How a policy that aims at reducing iatrogenesis favors drugs that are more likely to induce adverse reactions?  The statements in the second sentence are especially troubling. The authors should be cautious about such statements that need to be founded on published works.

At the end of the first paragraph, would not it be more appropriate to talk about precarious people than vulnerable people? Vulnerable people are also pregnant women and elderly people who might not be in a financial precarious status. By using precarious in place of vulnerable it would probably be more accurate and help the transition with the next paragraph where the authors should probably start with the definition a financial precarity in France.

The authors should be careful in Lines 95 to 98,

“However, to the best of our knowledge, no pharmaco-epidemiological study has so far examined the assumption that the improvement of the precarious populations’ health correlates with their free access to healthcare.  For that, we designed…”

These two sentences make believe that the object of the research is to assess this correlation which is untrue. The authors have no mean to assess precarious health status.

In the Methods:

In the statistical analyses section, lines 194 to 196 are redundant with lines 260 to 263 of the discussion. The presentation of the study limit and bias are better exposed in the discussion section.

In the Discussion:

Lines 272 to 278 are redundant with lines 220 to 223 the results section.

In the Discussion and Conclusion:

The idea that practitioners’ relationships with precarious patients might be a determinant of health care consumption is interesting but it is strange that the authors do not mention the precarious patient’s state of mind and behavior. It is most likely that some precarious people do not seek care although free because of shame, fatalistic state of mind or unawareness.

Author Response

Comments and Suggestions for Authors

The quality of the manuscript entitled “The Characteristics of Care Provided to Population(s) in Precarious Situations in 2015. A Preliminary Study on the Universal Health Cover System in France” has significantly been improved in since the first version. However, some issues remain:

Reply: We sincerely thank reviewer 1 for his/her positive evaluation of our work. Below, we address all queries in a detailed point-by-point response. The parts of the text that have been amended are indicated in red with the corresponding lines in brackets. We thank again reviewer 1 for his/her new comments and we hope that our new modifications have improved the general quality of our manuscript.

In the Abstract:

In the findings the authors state “6 out of 20 molecules were significantly under-reimbursed in precarious populations. 2 were over reimbursed.” This add up to 8 so how come the authors talk about the remaining 10 and not 12? “The 10 remaining molecules did not differ between groups.”

Reply: Thanks for this observation. This is a typo error. We have modified the abstract accordingly (please, see line 31).

In the Introduction:

The 2 first sentences are inconsistent. How a policy that aims at reducing iatrogenesis favors drugs that are more likely to induce adverse reactions?  The statements in the second sentence are especially troubling. The authors should be cautious about such statements that need to be founded on published works.

Reply: Many thanks for this very important comment. It is true that the two first sentences are actually inconsistent. We acknowledge that a clear distinction between the theoretical and practical dimensions of the economic models-based policies was lacking. In fact, theoretically, these economic models aim to improve efficacy, to reduce overconsumption and iatrogenesis etc. But practically, these very same models lead to limit the patients’ access to necessary drugs and to favour the use of less efficacious drugs or drugs with adverse reactions etc. We have modified these two sentences, mentioning this discrepancy between the models’ theoretical and practical dimensions. Moreover, we have added a reference to a published work, as requested. Please, see lines 48-52.

At the end of the first paragraph, would not it be more appropriate to talk about precarious people than vulnerable people? Vulnerable people are also pregnant women and elderly people who might not be in a financial precarious status. By using precarious in place of vulnerable it would probably be more accurate and help the transition with the next paragraph where the authors should probably start with the definition a financial precarity in France.

Reply: Thanks for this observation. It is true that the adjective “vulnerable” was not accurate enough.  We have replaced it with “precarious” as recommended. We have also modified the order of presentation in the next paragraph. We now start with the definition of the financial resources of precarious people in France and continue with that of non-precarious people. Please, see line 56 and lines 57-60.

The authors should be careful in Lines 95 to 98: “However, to the best of our knowledge, no pharmaco-epidemiological study has so far examined the assumption that the improvement of the precarious populations’ health correlates with their free access to healthcare.  For that, we designed…”

These two sentences make believe that the object of the research is to assess this correlation which is untrue. The authors have no mean to assess precarious health status.

Reply: Thanks for this comment. We have modified this part of the introduction and we have rewritten it more carefully. Please, see lines 99-102.

In the Methods:

In the statistical analyses section, lines 194 to 196 are redundant with lines 260 to 263 of the discussion. The presentation of the study limit and bias are better exposed in the discussion section.

Reply: Thanks for this comment. Our first aim was to indicate already in the methods section the reasons why we were not able to perform stratifications test with age, gender and health status as variables. It is true that the limits and biases of a study in general are better described and discussed in the discussion than in the methods section. Accordingly, we have removed this part of the text from the methods. The lack of stratification test is now only mentioned and discussed in the discussion, as previously. Please, see lines 270-273.

In the Discussion:

Lines 272 to 278 are redundant with lines 220 to 223 the results section.

Reply: Thanks for this comment. We have removed this part of the discussion that was redundant with the results sections. We only have brought indications concerning the ATC classes relative to molecules in brackets in the results sections. Please, see lines 224-233.

In the Discussion and Conclusion:

The idea that practitioners’ relationships with precarious patients might be a determinant of health care consumption is interesting but it is strange that the authors do not mention the precarious patient’s state of mind and behavior. It is most likely that some precarious people do not seek care although free because of shame, fatalistic state of mind or unawareness.

Reply: Many thanks for this comment. Actually, there was a short section in a previous version of the discussion that was devoted to this specific point (i.e., some precarious people do not seek care because of awareness disorders). We had removed this part, thinking that it was perhaps beyond the scope of the present manuscript. In fact, lack of insight in psychiatric conditions is one of our research topics. Accordingly, we have added a short statement relative to this hypothesis in the second revision of our manuscript. This has been done in the discussion but not in the conclusion because the latter focuses on our studies in progress (i.e., on GP’s burnout and empathy decrease). Please, see lines 396-404.

Reviewer 2 Report

When compared to the previous version, the manuscript has been greatly improved and the authors have answered my queries. I have some minor comment on wordings:

Line 54: “people with a monthly income ‘above/ higher than’ 720 Euros…” should be better while it sounds a bit odd to use “superior to”. Similarly, It is better to use “monthly income ‘below/ lower than (instead of is inferior to)’ this threshold (line 56)”.

Author Response

Comments and Suggestions for Authors

When compared to the previous version, the manuscript has been greatly improved and the authors have answered my queries. I have some minor comment on wordings:

Reply: We sincerely thank reviewer 2 for his/her positive evaluation. We are very glad that he/she considers our manuscript greatly improved.

Line 54: “people with a monthly income ‘above/ higher than’ 720 Euros…” should be better while it sounds a bit odd to use “superior to”. Similarly, It is better to use “monthly income ‘below/ lower than (instead of is inferior to)’ this threshold (line 56)”.

Reply: Thanks for this observation. We have modified the text (please, see lines 57-60).

This manuscript is a resubmission of an earlier submission. The following is a list of the peer review reports and author responses from that submission.

Round 1

Reviewer 1 Report

The manuscript entitled “The Characteristics of Care Provided to Population(s) 2 in Precarious Situations in 2015. A Preliminary Study 3 on the Universal Health Cover System in France” presents a social-economical descriptive analysis in an attempt to compare drug reimbursements according to patient health insurance coverage status.

MAJOR

The overall idea is interesting but the paper lack of clarity in every section and it is unclear whether the analyses answer the question and hypothesis formulated by the authors.

(1) are prescriptions of medication independent from the precarious situation reflected by the attribution of CMU-C status?

(2) The hypothesis was that free health cover does not rectify inequalities in prescriptions in precarious populations.

Introduction:

  • The presentation of what is CMU-C compare to CMU should be done. It could be very confusing for non-French readers who not aware of the health insurance system.The confusion is emphasized with a misleading sentence in the materials and methods section where it seems that CMU-C beneficiary are only reimbursed by 35%! But it is 65+35% ie. 100%, isn’t it?

“Our ISPL data provided only amounts reimbursed per region, per specialty 74 (medical specialties, midwives and healthcare units) and per type of social cover (reference healthcare cover (65% reimbursement), CMU-C (35% reimbursement), medical cover for illegal immigrants (Aide Médicale de l’état–_AME) (100% reimbursement), Alsace-Lorraine healthcare cover (90% reimbursement), and allowances for complementary health cover (Aide à la Complémentaire Santé ACS) (35% reimbursement)). “

The authors should explain that people with the reference healthcare cover of 65% and no special social cover should either pay the rest (co-pay) or can sign up for an additional private insurance.

  • It would be more comprehensive for the international reader to define 2 groups such as “precarious” versus “non-precarious” or “social help beneficiaries” and “general beneficiaries”.

Materials and Methods:

Data collection and analysis

As previously highlight for the introduction, the section on Data collection and analysis is confusing because of abbreviation.

  • For the prescriber choice, although strangely presented, we can understand the reference to midwives and gender relation, why not other specialists?  
  • For the description of the ATC analysis a practical example with 1 drug should be presented instead if using theoretical explanation LCCLLCC.
  • Line 94 “Data was extracted from the ISPL database, with analyses using Statistica Software” should be at the end of the statistical analysis section

Statistical analysis section:

  • Does the abbreviation DDD in table 2 refers to “daily doses delivered” of line 114? Line 114, a clear explanation with formula would greatly help the readers to know what is the level of reimbursement.
  • The authors stated that the “daily doses delivered” were not computed but collected from the database per insured subject over 20 years of age.
  • How many subjects does that correspond to in each group (ref and CMU-C)?
  • The DDD presented in Table 2 are average? It should be explained in the method.
  • How the authors justify the use of t-test? Was it unilateral or bilateral test?

Results:

Table 1 is not really informative regarding the questions and hypothesis.

Table 2 is difficult to understand as it lacks of details on the variable displayed:

  • In table2, does the “DDD ref” correspond to the 65% coverage (rest being copay or private insurance) and the “DDD CMU” correspond to the “CMU 65%+35% CMU-C” (i.e. free care)? In other words, the overall expenditure for the national health insurance system. As proposed above, a clear definition of the 2 groups (precarious and non-precarious) given in introduction or in the materials and methods would clarify the table.
  • The ATC class level 3 or 4 of the listed drug could be informative to discuss the possible disparities observed between groups of patients.
  • Line 124: the 16.3% is different from the 17,5% of Table 2. What number is correct?
  • Line 128: (3.1 versus 53.7%) should not it be from 3.1 to 53.7%?
  • Line 128: for ivermectine the authors wrote in the text that it overprescribed but the test result is non-significant (NS) in Table 2. What is true?
  • Overall, it is unclear how less reimbursement is synonymous of less prescription. Is it not a rough shortcut?
  • Overall, are the comparisons of DDD adjusted for the difference of prevalence of diseases in the 2 groups?

Discussions:

  • Line 196: again, the authors conclude on under-prescription of 2 drugs for which the statistical test was non-significant.
  • The discussion will have to be reviewed according to the revision of the methods and results section.
  • The authors should also note as a limit in the discussion section the poor quality of the CMU-C variable in the Health Insurance Information Systems (SNDS) and the other limit of their dataset.

English should be reviewed:

Several typos and formulation are to be corrected. I suggest that the manuscript be reviewed by a native English or professional translator.

“…obtained aggregated data regarding reimbursements from the [freanch]  [healt] care system and [insureance] providers…”

MINOR:

In table2 why some p-value (not p student) are in bold? Why some drug names start with a capital letter and some do not?

Author Response

nnn

Reviewer 2 Report

This is a retrospective observational analysis, which aimed to evaluate the impact of CMU by comparing the average cost reimbursed per patient per year of the 20 most prescribed drug use among the population under CMU and that of the general population in 2015. With Student’s t test, it is found that DDD per patient for econazole, ciclopirox, ivermectine and chromoglicique acide was higher for the CMU group, when compared to the reference group. On the other hand, per patient DDD for metformin, acetyl salycyclic acid, atorvastatin, rosuvastatin, serenoa repens, tamzulosin, amoxicillin, ibuprofen, paracetamol, and timolol were lower. CMU did not cancel out differences in drug prescription for the precarious populations, where partly was due to the epidemiological and non-clinical factors.  

As a known social determinant of health, equitable access to health is essential for everyone. The authors of this study raised a very good research question and it is important to evaluate whether the complementary universal health cover system through CMU improved the situation of poor and disadvantageous individuals in France. Yet, further corrections and editing are required before the paper is ready for publication.

  1. The background of the French health insurance system is not clear. In section 2.2, it consists of four schemes at least: reference health cover (65% reimbursement), CMU-C (35% reimbursement), ACS (100% reimbursement); Alscace-Lorraine (90% reimbursement). It seems that the reimbursement rate of CMU-C is the lowest among the other four. This indicates members of CMU-C need to co-pay 65% of health cost. Can the authors claim the scheme helps the precarious population in this situation?
  2. In line 86, the poverty threshold is not clear: Is it 1000 Euro per week/ per fortnight/ per month?
  3. In line 107, it is noted that “ In each ATC class, the second-level class entailing the most expenditure was selected” So, the breakdown of yearly reimbursement claims and reimbursement expenditure in Table 1 should be based on level 2 ATC classification, not level 1.
  4. In line 121, it is noted that “… three (H, N and P) presented a molecule that cumulated the largest number of reimbursed packs and the largest reimbursed amounts.” However, in Table 1, it is ATC classes A, C, and N was the highest.
  5. In Table 1, please provide the content of each ATC classification: A - Alimentary tract and metabolism; B - Blood and blood-forming organs; C - Cardiovascular system; ... It is difficult to comprehend this table by showing the first digit of ATC classification.
  6. In Table 2, it shows the top 20 molecules with the highest reimbursed amounts. Without any explanations, I do not understand the term “DDD” – Is it the average cost per patient per year (as noted in the abstract)? Or average reimbursement amount per patient? As a research article, we cannot make assumptions on any terms that are uncertain.
  7. Following the question 6: As the two insurance schemes have different reimbursement rate (CMU:35%) and (reference health cover:65%), I am doubted if it is appropriate to compare the average reimbursement per patient.
  8. In Table 2, the column “CMU/ref” is calculated by dividing DDD CMU with DDD ref. Is it correct? What is the purpose? As Student’s t-test was used, the authors should show the absolute difference (i.e. DDD CMU - DDD ref), rather than relative difference.
  9. I think the results section should be lengthened and mentioned any key results that will be discussed in the discussion section
  10. The explanations noted in the discussion section highlights the main problem of this study. This study is not a randomized trial, which controls patients’ demographic and disease profile (including the type of disease diagnosed and the severity of disease at baseline. So, the differences can be attributed to other factors, instead of the policy. This has to be adjusted by either matching, stratification or modelling in the analysis. Otherwise, what is the value of this analysis?  

Author Response

Reviewer 2

Open Review

Comments and Suggestions for Authors

This is a retrospective observational analysis, which aimed to evaluate the impact of CMU by comparing the average cost reimbursed per patient per year of the 20 most prescribed drug use among the population under CMU and that of the general population in 2015. With Student’s t test, it is found that DDD per patient for econazole, ciclopirox, ivermectine and chromoglicique acide was higher for the CMU group, when compared to the reference group. On the other hand, per patient DDD for metformin, acetyl salycyclic acid, atorvastatin, rosuvastatin, serenoa repens, tamzulosin, amoxicillin, ibuprofen, paracetamol, and timolol were lower. CMU did not cancel out differences in drug prescription for the precarious populations, where partly was due to the epidemiological and non-clinical factors.  

As a known social determinant of health, equitable access to health is essential for everyone. The authors of this study raised a very good research question and it is important to evaluate whether the complementary universal health cover system through CMU improved the situation of poor and disadvantageous individuals in France. Yet, further corrections and editing are required before the paper is ready for publication.

Reply: We thank sincerely reviewer 2 for taking time to evaluate our work and for his/her valuable comments. Below, we address all queries in a detailed point-by-point response. We have amended most parts of our initial text. We hope that our modifications have clarified and improved our manuscript.

1. The background of the French health insurance system is not clear. In section 2.2, it consists of four schemes at least: reference health cover (65% reimbursement), CMU-C (35% reimbursement), ACS (100% reimbursement); Alscace-Lorraine (90% reimbursement). It seems that the reimbursement rate of CMU-C is the lowest among the other four. This indicates members of CMU-C need to co-pay 65% of health cost. Can the authors claim the scheme helps the precarious population in this situation?

Reply: Many thanks for this very important comment. We agree that our description of the French Health Insurance System in the original version of our manuscript was not clear, leading to misunderstanding, and needed to be significantly improved. Based on the comments of reviewer 2, we have clarified, in the revised version of our manuscript, the sections devoted to the description of the populations on which the present study focuses, i.e., the introduction (please, see lines 57-72) and methods sections (please, see lines 133-145). We hope that our text is now less confusing and more understandable.

 There are four different types of health covers in France:

  1. The General Health Cover (in French: “Regime Général d’Assurance Maladie” or RGAM).
  2. The Alsace Moselle Medical Cover (in French: “Régime Alsace-Moselle” or RAM).
  3. The Universal Medical Cover (in French: “Couverture Médicale Universelle” or CMU)
  4. The State Medical Aid (in French: “Aide Médicale d’Etat” or AME)

Populations benefiting from the RGAM or RAM are non-precarious populations. The criterion used to define a population as non-precarious is their monthly income. When this is superior to 720 Euros for a single person and superior to 1080 Euros for a couple living in an urban area, these people are considered “non-precarious”. The patients benefiting from the RGAM are 65% reimbursed for their medical expenses (consultations, pharmacological treatments, hospitalizations, exams etc.). Thus, to be fully reimbursed, these have either to pay the 35% remaining costs (or co-pay) or to sign up an additional private insurance. This concerns about 91% of the French population. RAM has been settled in the immediate post-World War II period. It concerns a very low part of the French population and does not correspond to the standards laid down in the General Health Cover. For this reason, data from RAM-beneficiaries were not included in the present study.

In contrast, people with a monthly income inferior to 720 Euros (single person) or 1080 Euros (couple) are considered “precarious”. Populations responding to these financial resources criteria benefit from the CMU. In this case, CMU-beneficiaries are reimbursed up to 65% of their medical expenses by the RGAM, comparably to non-precarious people. However, their limited financial resources allow them neither to pay the 35% remaining costs nor to take out an additional private insurance in contrast to non-precarious patients. Consequently, the French Health Insurance System takes care of the 35% additional costs for CMU-beneficiaries to be finally 100% reimbursed for their medical expenses (i.e., 65% + 35%). Please, note that we have simplified our text and did not refer to CMU-C anymore, which was too complicated to non-French readers and was too detailed to the scope of the present study.

AME only concerns irregular migrants. As such, it does not target precarious populations per se even if most foreigners in irregular situations experience precarious life conditions. For this reason, data from AME-beneficiaries were not included in the present study.

Our study, thus, focused on CMU-beneficiaries (corresponding to the precarious group) and on RGAM-beneficiaries (corresponding to the non-precarious group).

2.In line 86, the poverty threshold is not clear: Is it 1000 Euro per week/ per fortnight/ per month?

Reply: Many thanks for this comment. The threshold determining poverty is equal to 720 Euros for a single person and to 1080 Euros for a couple (living in an urban area) per month. This is now specified in the revised version of our manuscript (please, see Introduction, lines 57-60).3.

3.In line 107, it is noted that “ In each ATC class, the second-level class entailing the most expenditure was selected” So, the breakdown of yearly reimbursement claims and reimbursement expenditure in Table 1 should be based on level 2 ATC classification, not level 1.

Reply: Many thanks for this comment. please, see line 176-8

4. In line 121, it is noted that “… three (H, N and P) presented a molecule that cumulated the largest number of reimbursed packs and the largest reimbursed amounts.” However, in Table 1, it is ATC classes A, C, and N was the highest.

Reply: Many thanks for this comment. please, see line 176-

5. In Table 1, please provide the content of each ATC classification: A - Alimentary tract and metabolism; B - Blood and blood-forming organs; C - Cardiovascular system; ... It is difficult to comprehend this table by showing the first digit of ATC classification.

Reply: Many thanks for this remark. We acknowledge that the presentation of table 1 was not clear enough in its previous state. We have added the content of each AT classification, as recommended by Reviewer 2. Please, see page 10. We also clarified ATC classification in the Methods section (please, see lines 148-161). The Table 1 is now more comprehensible thanks to the Methods

6. In Table 2, it shows the top 20 molecules with the highest reimbursed amounts. Without any explanations, I do not understand the term “DDD” – Is it the average cost per patient per year (as noted in the abstract)? Or average reimbursement amount per patient? As a research article, we cannot make assumptions on any terms that are uncertain.

Reply: Many thanks for this comment. We agree that the presentation of table 2 needed to be improved. In fact, drugs consumption can be expressed in cost, number of units, number of prescriptions or in the drugs physical quantity. Here, we used a technical unit measurement which is termed the “Defined Daily Dose” or DDD. DDD corresponds to the assumed average maintenance dose per day for a given drug and according to its main indications in adults. DDD are only assigned for medicines with ATC codes. In general, applying DDD enables to examine changes in drug utilization over time, to compare and evaluate the effect of an intervention on drug use, to document the relative therapy intensity with various groups of drugs, to track changes in the use of a class of drugs and to evaluate regulatory effects and effects of interventions on prescribing patterns.

Accordingly, we have added in the revised version of our manuscript the definition of DDD in the caption of Table 2 (please, see page 11). We now also clearly define DDD in the methods section (please, see lines 164-186). We hope that these modifications render our text more readable and understandable.

7. Following the question 6: As the two insurance schemes have different reimbursement rate (CMU:35%) and (reference health cover:65%), I am doubted if it is appropriate to compare the average reimbursement per patient.

Reply: Many thanks for this very important comment. For detailed explanations, please, see our reply to the first query above.

8. In Table 2, the column “CMU/ref” is calculated by dividing DDD CMU with DDD ref. Is it correct? What is the purpose? As Student’s t-test was used, the authors should show the absolute difference (i.e. DDD CMU - DDD ref), rather than relative difference.

Reply: In the revised version of our manuscript and based on the comments of Reviewer 1, we have re-analysed our data using non-parametric tests, i.e., Mann-Whitney’s tests. Firstly, we tested the normality for each molecule (precarious vs. non-precarious) using Shapiro-Wilk tests. For each molecule, the p-value was inferior to 0.05, indicating that the data were not normally distributed. Secondly and based on these normality tests, we calculated Mann-Whitney’s test for each molecule.  Accordingly, we have rewritten the methods section describing the used statistical analyses, (please, see lines 189-198), the results section (as our results were slightly modified) (please, see lines 201-221) and the entire discussion. We have also modified the table 2 according to the comments of Reviewer 2.

9. I think the results section should be lengthened and mentioned any key results that will be discussed in the discussion section

Reply: Thanks for this remark. We have lengthened, although only very slightly, the results section and highlighted the key results, i.e., the results that are discussed in the Discussion section. We hope that these modifications have made the presentation of the main results more directly understandable.

10. The explanations noted in the discussion section highlights the main problem of this study. This study is not a randomized trial, which controls patients’ demographic and disease profile (including the type of disease diagnosed and the severity of disease at baseline. So, the differences can be attributed to other factors, instead of the policy. This has to be adjusted by either matching, stratification or modelling in the analysis. Otherwise, what is the value of this analysis?  

Reply: Many thanks. We agree with this fundamental criticism. The aim of the present study was to examine the way precarious and non-precarious populations are reimbursed for drugs that are prescribed by their GPs. For that, we obtained permission to contact the French Health Insurance System. It is worth noting that this permission is only rarely given. People in charge within the French Health Insurance System accepted to provide us with the requested data (i.e., DDD). This is also an exceptional situation. However, at this very first stage, we were not allowed to collect the patients’ data, i.e., the data relative to the patients’ age, gender, underlying medical conditions etc. Accordingly, we were not able to adjust our statistical analyses to the above mentioned variables of age, gender, diagnosis, severity etc. Thus, we were not able to use ponderation or stratification tests (etc.) (confusion biases). We acknowledge that this lack of information is an important limitation to our study. This limitation is now clearly discussed (please, see lines 244-263). However, we here would like to insist that (1) this study is the first step of a larger research project, (2) that our first results raise important questions regarding the French Health Insurance System that need to be addressed, criticized and discussed and (3) that this first study – although still preliminary – will significantly help us (a) to collect further data and information relative to the precarious and non-precarious populations and (b) to relate these data to the data relative to the prescriptions. Furthermore, we also believe that the data relative to the practitioners are also crucial, i.e., the age, gender, mental and/or psychiatric conditions etc variables. As an example and based on our previous results (see Thirioux et al., 2016; Thirioux et al., 2020), we are now investigating whether practitioners who care precarious populations suffer more significantly from burnout syndrome than practitioners caring non-precarious populations and, if correct, how this may negatively impact their drugs prescription. This ongoing study is also mentioned in the revised version of our manuscript (please, see lines 386-419).  
